# Immigrating and vicinity are not risk factors in the prevalence and transmission rate of human T-lymphotropic virus type 1: A Survey in an endemic region of Iran and Afghan refugees

Maryam Mahdifar[1], Mohammad Reza Akbari-Eidgahi[2], Arman Mosavat[3], Alireza Pourreza[1], Mohammad Mehdi Akbarin[1], Narges Valizadeh[1], Seyed Abdolrahim Rezaee[1]*, Houshang Rafatpanah[1]*

1 Inflammation and Inflammatory Diseases Division, Immunology Research Center, Mashhad University of Medical Sciences, Mashhad, Iran, 2 Biotechnology Research Center, Semnan University of Medical Sciences, Semnan, Iran, 3 Blood Borne Infections Research Center, Academic Center for Education, Culture, and Research (ACECR), Razavi Khorasan, Mashhad, Iran

* RezaeeR@mums.ac.ir (SAR); RafatPanahH@mums.ac.ir (HR)

## Abstract

Human T-lymphotropic virus type 1 (HTLV-1) is a retrovirus associated with two life-threatening diseases; HAM/TSP and ATLL. Due to the slow-growing HTLV-1 infection worldwide, WHO urged for elimination. A large border with Afghanistan, northeast Iran is an endemic region for HTLV-1 infection. Historically, Afghanistan has common sociocultural similarities to Persian peoples. This study was conducted to evaluate HTLV-1 prevalence in Afghan refugees. Also, the HTLV-1 transmission rate and understanding of whether or not the Silk Road has been the route of HTLV-1 infection to Iran were investigated. This case-control study was conducted in a rural area of Fariman city, with Afghan residents who migrated around 165 years ago, from 1857, the Treaty of Paris at the end of the Anglo-Persian war, and a refugee camp in Torbat-e-Jam city. These populations in HTLV-1 endemic area were compared to a segregated population of Afghan refugees in Semnan, the centre of Iran. Blood samples of 983 volunteers were assessed with the ELISA method for the presence of HTLV-1 antibodies and then confirmed by PCR technique. All samples from Afghan refugee camps, Semnan and Torbat-e-Jam, were negative for HTLV-1 infection. However, the prevalence of HTLV-1 infection in Fariman, a rural population of Afghan origin, was approximately 2.73%. The results showed that HTLV-1 is not endemic in Afghanistan, a war-stricken region with refugees distributed worldwide. The land Silk Road has not been the route of HTLV-1 transmission to Northeastern Iran. Importantly, HTLV-1 endemicity might occur during a long time of living in an endemic area.

**Data Availability Statement:** All datasets generated or analysed during the current study are included in this paper.

**Funding:** This study was financially supported by the Vice-Chancellor for Research and Technology, Mashhad University of Medical Sciences, Mashhad, Iran [grant no. MUMS 88259, recipient: SAR] and the Vice-Chancellor for Research and Technology, Semnan University of Medical Sciences, Semnan, Iran [grant no. SUMS 88259, recipient: MRAE. The funders had no role in study design, data collection and analysis, decision to publish, or preparation of the manuscript.

**Competing interests:** The authors have declared that no competing interests exist.

## Introduction

The human T-lymphotropic virus type 1 (HTLV-1) is a type C retrovirus belonging to the *Retroviridae* family, which is associated with two significant diseases; HTLV-1-associated myelopathy/tropical spastic paraparesis (HAM/TSP) and adult T-cell leukaemia/lymphoma (ATLL) [1].

The geographic distribution, which has a high prevalence of HTLV-1, is restricted to distinct regions, including Southwestern Japan, Central, and South America, the Caribbean islands, some parts of Africa, Europe, and Australia-Melanesia [2]. In Iran, over five provinces [3,4], notably Razavi and North Khorasan provinces (around 7 million populations, Iranian Census 2016), have documented HTLV-1 infection [5,6]. Previously, Mashhad (2.1%), Sabzevar (1.6%) and Neyshabour (3.7%), Kalaleh (2.6%), and Ali-Abad (1.9%) were identified as the most critical endemic areas in the Razavi Khorasan and Golestan provinces, respectively [4–7]. It is not clear how the virus was first introduced in Northeastern Iran. However, Safai *et al.* proposed some possible routes of HTLV-1 transmission to Razavi Khorasan in Iran. The main possible route was the Silk Road from the regions where HTLV-1 infection was more prevalent [8]. It is well known that the ancient Silk Road has been a trading route for several centuries in the past, serving as the primary connection between Europe and Asia. The route traverses a central geographical region during human expansion from Africa [8,9]. However, our recent phylodynamic study suggested that the transmission of HTLV-1 to Iran was during the Mongolian invasion of Persia, China, and parts of Japan [10].

Historically, Iran and Afghanistan had been part of the Great Persia in the Silk Road route. Furthermore, a large population of Afghans has settled down in East Khorasan, mainly in Razavi Khorasan, in Iran for centuries [11]. Nevertheless, most Afghans immigrated to Iran (more than 4 million) from the beginning of the civil war in 1979 as immigrants, asylums, or job seekers [12].

The present study was conducted to find, firstly, for a virus with cell-to-cell spreading, not free particles such as HTLV-1 [13], whether or not immigration and neighbourhoods are risk factors for endemic ties as it is not still clear how the virus enters this endemic region [10]. Secondly, what was the role of the land route of the Silk Road in such spreading? Thirdly, is HTLV-1 prevalent in different Afghan refugees ethnic groups living in Iran? Fourthly, the effect of living time in an endemic region on HTLV-1 prevalence. Therefore, the present study evaluated the prevalence of HTLV-1 infection in three segregated populations of Afghans living in Iran in comparison to the endemic region.

## Materials and methods

### Ethics approval and population setting

The Afghanistan consulate in Mashhad approved this study and then coordinated sampling with the Governors of Razavi Khorasan and Semnan provinces. This study protocol was reviewed, approved, and supervised by the Biomedical Research Ethics Committee of the Mashhad University of Medical Sciences [**IR.MUMS.REC.88259**]. The written informed consent forms were obtained and signed by all the participants. All methods were performed following relevant guidelines and regulations.

A total number of 983 blood samples from random volunteers were collected and divided into three groups. The first group included 256 Khawari, a tribe in Afghanistan who immigrated and settled down in Iran at the end of the Anglo-Persian war in 1857 [11], living in rural areas around Fariman city. Fariman is located in Razavi Khorasan, the main HTLV-1 endemic region of Iran. In the second group, the HTLV-1 prevalence was conducted in

Afghan refugee camps located in Semnan province in the centre of Iran. From all camp areas, 424 individuals were selected by multi-stage cluster sampling. The third group included 303 Afghan job seekers in a refugee camp in Torbat-e-Jam working in Mashhad, the capital of Razavi Khorasan, in northeast Iran. A questionnaire was provided for the data collection on demographic, clinical information, and the risk factors for HTLV-1 infection.

### HTLV-1 serologic assay and PCR analysis

Five mL of the venous blood sample was taken from each individual. According to the manufacturer's instructions, the serum was separated, and DNA was extracted by a commercial kit (Genet Bio, South Korea). Enzyme-linked immunosorbent assay (ELISA; Dia. Pro Diagnostic, Italy) was carried out to screen anti-HTLV-1 antibodies in serum. Reactive ELISA samples were then applied to a conventional polymerase chain reaction (PCR), as previously described in detail [6], to confirm infection.

### Statistical analysis

All data were analysed using SPSS software ver.11.5 (SPSS, Chicago, IL). Descriptive statistic was performed, and the results were presented as frequency, percentage, and mean±SD criteria.

## Results

The mean age of the study population was 36±12, 39.3±16, and 27.62±0.8 in Fariman, Semnan, and Torbat-e-Jam, respectively. In the Fariman study group, 205 (80%) out of 256 subjects were male, and 51 (20%) were female. 217 out of 424 subjects from Semnan were female (51.2%), 207 (48.8%) were male, and all subjects in Torbat-e-Jam's camp were male. Table 1 shows the age distribution among the studied populations.

All possible pathways involved in the transmission of the virus, such as breastfeeding, blood transfusion, surgery history, unprotected sexual intercourse, and drug injection, were considered. Among Afghans living in refugee camps, most individuals involved in the study (85%) have had a history of breastfeeding, 12% had a history of surgery, five subjects were blood transfusion during hospitalisation, and one case had a history of drug use. Sexual activities are taboo among this study population. It was not included in the questionnaire. Furthermore, ethnic origin is also taken to account. Fig 1, shows the birthplace-based frequency of Afghans.

### Ethnicity frequency among Afghan study groups

The study population included five major ethnic groups; Baluch, Hazara, Pashtun, Tajik, and Uzbek. The major ethnic group in the Semnan refugee camp population was Baluch (172/40.5%), and the major ethnic group in the Torbat-e-Jam refugee camp was Tajik (99/32.6%). The second most common ethnic group was Uzbeks, with 113 (26.6%), and Hazaras, 59 (19.47%) in Semnan and Torbat-e-Jam. Arabs had 5.6% and 2.31% frequencies in Semnan and

**Table 1. Age-based distribution of individuals.**

| Group | Number | Mean age | | Gender | |
|---|---|---|---|---|---|
| | | | Male | | Female |
| Fariman | 256 | 36±12 | 205 (80%) | | 51 (20%) |
| Semnan | 424 | 39.3±16 | 207 (48.8%) | | 217 (51.2%) |
| Torbat-e-Jam | 303 | 27.62±0.8 | 303 (100%) | | - |

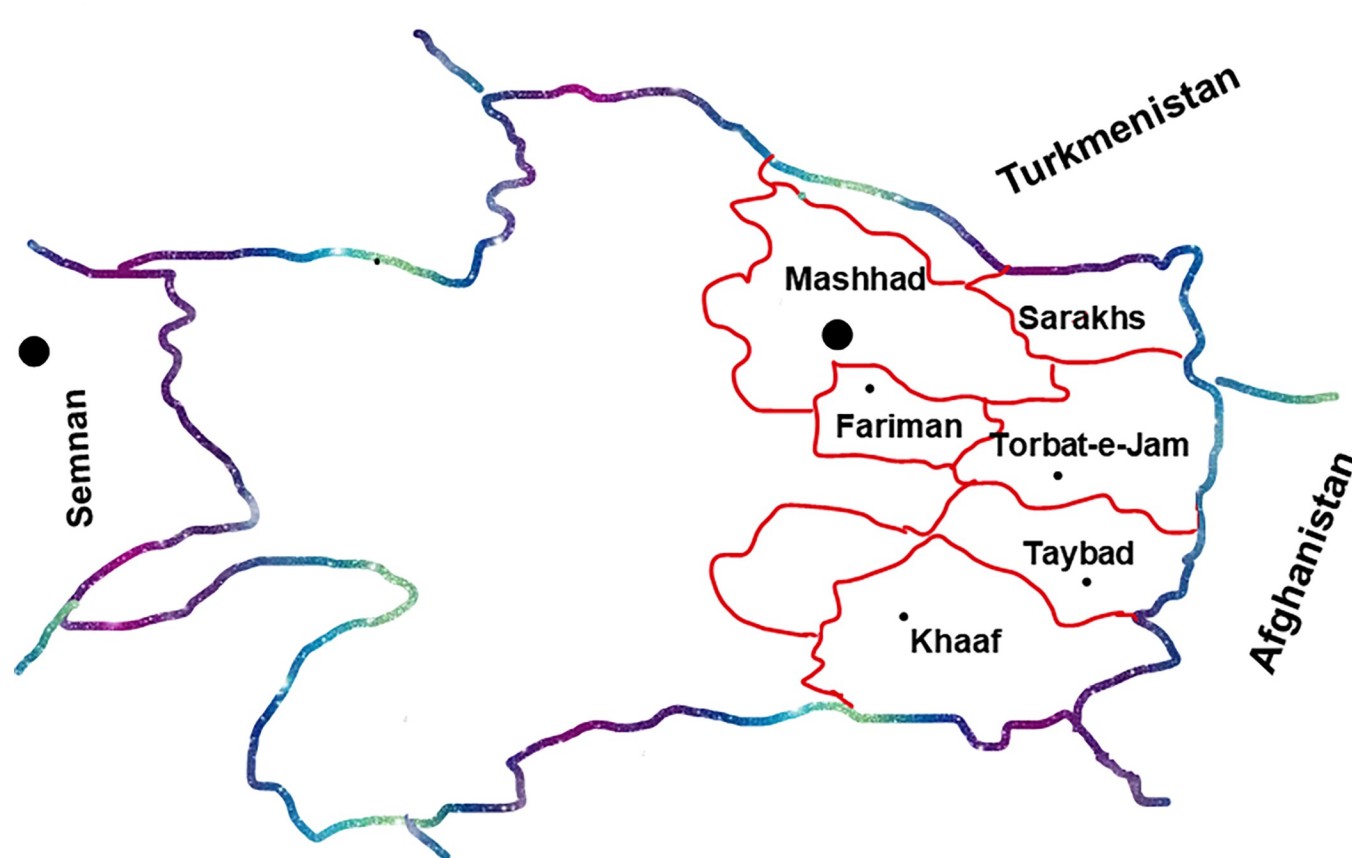

**Fig 1. Birth place-based frequency of Afghans studies population.** Eastern cities of Razavi Khorasan, the map shows the main locations of studied cities and the neighbourhoods. The Razavi Khorasan map was translated and depicted according to the National Cartographic Center of Iran; 2021, (https://www.ncc. gov.ir/images/docs/files/000005/nf00005286-2.pdf).

Torbat-e-Jam. The other ethnic groups included Aimak, Pashai, and Nuristani. However, in Fariman, the major ethnic group was Khawari (the stable Afghans residence in Razavi Khorasan since 1857) (34%). Table 2 shows the frequency of the principal ethnic groups.

## HTLV-1 serological findings

The serological results disclosed that all samples from the two refugee camps were negative for HTLV-1 antibody. To confirm that all the samples were negative for HTLV-1, PCR was applied to re-examine the samples. Further, the results showed no positive values for the samples. However, in the participants from a rural area of Fariman city, anti-HTLV-1 antibodies were detected in serum samples of five cases out of 256 (2.73%). Also, positive samples were confirmed by a conventional PCR method.

Table 2. Main ethnic groups among Afghan study groups.

| Study groups | Pashtun n (%) | Hazara n (%) | Tajik n (%) | Uzbek n (%) | Baluch n (%) | Arab n (%) |
|---|---|---|---|---|---|---|
| Semnan Afghan population | 96 (22.6) | 13 (30) | 6 (1.41) | 113 (26.6) | 172 (40.5) | 24 (5.6) |
| Torbat-e-Jam Afghan population | 51 (16.8) | 59 (19.47) | 99 (32.67) | 50 (16.5) | 37 (12.21) | 7 (2.31) |

According to the Britannica Encyclopedia, under the title of "Silk Road; Trade Route", this longest ancient trade route from China to Europe has at least two land routes and one sea road, which all reach the main road in Mashhad, which became the capital of the Greater Khorasan in 1729. The road originated from Beijing or Shanghai in China and Anxi, branched off to Balkh or Bukhara, and finally crossed Mashhad. The sea route originated from the Bay of Bengal, crossed in Balkh and reached Mashhad; therefore, Mashhad has been the main centre of this road in the territory of Ancient Persia (https://www.britannica.com/topic/Silk-Road-trade-route). However, in the present study, there was not any apparent HTLV-1 infection in Afghans. Nevertheless, the land branches of the Silk Road were not the route of HTLV-1 spreading in Razavi Khorasan. In addition, our new study showed that the Mongolian invasion might be the time of HTLV-1 entry into Iran [10].

## Discussion

Southeast Japan has been reported as the first endemic area of HTLV-1 in Asia. Later, the preliminary reports of HTLV-1 infection were conducted in 1996 at Mashhad, northeast Iran, as the second endemic area of the virus in this continent [8]. Razavi Khorasan province, with a population of 6.2 million (Iranian census 2016) along the route of the Silk Road, is a tourist and pilgrimage region and shares a wide border with Afghanistan in the east and Turkmenistan in the north.

Although the prevalence of the virus in this province is around 2.3%, Neyshabour in a ~130 Km distance, has been identified as the most infected area (3.7–7%), which seems to be the source of the spreading of HTLV-1 in Iran [7,14].

Recently, a trace of infection has been found in other provinces close, such as Golestan or far from this endemic area (Urmia, West Azerbaijan of Iran) [15]. In contrast, the HTLV-1-associated diseases were referred to the HTLV-1 clinic, Ghaem Hospital, HTLV-1 Foundation, and Mashhad University of Medical Sciences from 19 provinces. This distribution pattern convinced the National Blood Transfusion Organization to conduct a survey and decided to screen the blood donors for HTLV-1 [14].

The prevalence of HTLV-1 infection and associated diseases in other parts of Asia seems very low [16,17]. It is more likely that this issue is due to the lack of sizeable molecular epidemiology studies in different countries of this continent.

However, the presence of HTLV-1 in the sporadic form in Middle East countries, including Kuwait and Israel, seems to be related to the Mashhad lineage of HTLV-1 [18–21].

Taken together to have a clue about the HTLV-1 spreading and the probable transmission rate, the present study was conducted. The prevalence of HTLV-1 infection was evaluated in Fariman (a city near Mashhad with an inhabitant of 34% Khawari, a tribe in Afghanistan who immigrated and settled down in Iran at the end of the Anglo-Persian war, 1857) and two Afghan refugee camps (one in Semnan as a non-endemic region, and the second, a city in Khorasan province, Torbat-e-Jam as an endemic area). These two Afghan populations immigrated during the civil war around 20 years ago.

The findings demonstrated no HTLV-1 infection in the studied population, neither in non-endemic nor in the endemic areas from Afghan refugee camps who frequently travelled between Iran and Afghanistan for more than 20 years. However, in Fariman, the prevalence of HTLV-1 infection was about 2.73%, similar to the capital of Razavi Khorasan, Mashhad.

Some Afghans living in Iran have settled in rural and urban areas around Fariman, located south of Mashhad. These Afghan immigrants integrated well into the Fariman and Khorasani communities [11] as they have lived for more than 165 years in this region of Iran.

According to these findings, two hypotheses could be suggested. Since Farimani's original citizens have been living for hundreds of years with Khorasanies, they were expected to be infected with this ancient virus.

Afghans have been in close contact with Farimanis and Khorasanies, too. In addition, there have been no positive cases in Afghans from Semnan or Torbat-e-Jam camps. Firstly, short-time contact by living in an endemic region for HTLV-1 did become a risk factor for infection. Secondly, as Afghanistan and Iran have been along the route of Silk Road, it could not be the route of spreading the HTLV-1 infection. It is more likely that Razavi Khorasan historical cities, Mashhad or Neyshabour, have been the source in Iran and maybe in the Middle East. However, the question remains unresolved: What is the infection source in the Khorasan provinces as a touristic and pilgrimage area? One possible assumption is the Sea route of the Silk Road from India. Some cases of HTLV-1 infection in India have been reported to have been genetically associated with HTLV-1 strains from Mashhadi Jews living in Israel (Nerurkar, V. R. *et al.*,) or perhaps Japan, as the main endemic area in Asia [22]. Therefore, HTLV-1 could be transmitted into Iran from India or Japan by the Silk Road sea route through trade routes or occupation (Fig 1).

Studies have shown that the prevalence of HTLV-1 in blood donors in the neighbouring countries of Iran, such as Pakistan, Turkmenistan, and Turkey, was low [16]. Southwest Asia is a non-endemic region of HTLV-1, and it has been reported that the seroprevalence rates of the virus are meagre among blood donors in Saudi Arabia. The phylogenetic analysis was similar to the HTLV-1 isolates from Mashhad [21]. A few cases of HTLV-1 carriers and HTLV-1-associated diseases have also been reported in Iraq [23]. According to these results, religion is not associated with spreading HTLV-1. Saudi Arabia is the leading pilgrimage country for Muslims, and Iraq is the main pilgrimage country for Shiites, which are not endemic and confirmed that a short time connection, even with an infected subject, could not be the risk factor for HTLV-1 spreading. Moreover, the rarely infected subjects had Khorasani origin.

In the last decade, a question has been raised about why the HTLV-1 prevalence is high in northeast Iran. One of the possible answers to this question was that the virus had been introduced to Iran via Silk Road, which passes through Afghanistan. However, no Afghan refugees living in camps near Semnan and Torbat-e-Jam were infected with HTLV-1. So far, due to social conditions and civil wars, HTLV-1 prevalence has not yet been investigated among blood donors and the general population in Afghanistan. Thus, there is no data available regarding this infection. However, a study by Husseini *et al.* investigating the prevalence of blood-borne viral infections among the general population in Afghanistan reported that the frequency of HTLV-1/2 was about 0.6%, although they suggested further monitoring HTLV-1/2 in Afghanistan to confirm their observation [24]. Studying the prevalence of blood-borne viruses among haemophilia patients in Afghanistan demonstrated that none were positive for HTLV-1 infection [25]. Haker *et al.* report the first case report of transmitted HTLV-1 infection in an American soldier who received fresh whole blood after injury in Jalalabad, Afghanistan, from a US-born 32-year-old white male donor. Forty-four days after transfusion, the soldier tested positive for HTLV-1. Serologic tests showed the blood donor was seropositive for HTLV-1, although the donor had shown no symptoms or signs [26].

Our recent phylogenetic and phylodynamic study on the samples from endemic areas suggested that multiple introductions have been involved in HTLV-1 spreading into Iran. Most of these spreading happened around the Mongol invasion from Iran to China and parts of Japan prior to the 15th century [10].

This study has some inevitable limitations; firstly, conducting the study in the chaos of war in the general population of Afghanistan was nearly impossible. Therefore, we had to conduct the study in two refugee camps in Iran with fewer confounding variables. Secondly, due to the

cultural and religious beliefs conservative population with special rules and regulations, we could not convince them to sample from more women.

## Conclusions

It seems that HTLV-1 infection is not endemic in Afghanistan. The land Silk Road has not been the route of HTLV-1 transmission to northeast Iran, perhaps sea Silk Road routes or during the Mongol invasion. Of importance, HTLV-1 endemicity might occur for a long time in peculiar immigrants living in an endemic area.

## Acknowledgments

Thanks to all the participants in the study, particularly our colleagues in Inflammation and Inflammatory Diseases Division, Immunology Research Center, MUMS, for their valuable help and support.

## Author Contributions

**Conceptualization:** Mohammad Mehdi Akbarin, Seyed Abdolrahim Rezaee, Houshang Rafatpanah.

**Data curation:** Mohammad Reza Akbari-Eidgahi, Arman Mosavat, Houshang Rafatpanah.

**Formal analysis:** Houshang Rafatpanah.

**Investigation:** Narges Valizadeh, Seyed Abdolrahim Rezaee.

**Methodology:** Alireza Pourreza, Seyed Abdolrahim Rezaee, Houshang Rafatpanah.

**Project administration:** Seyed Abdolrahim Rezaee.

**Resources:** Mohammad Reza Akbari-Eidgahi.

**Supervision:** Seyed Abdolrahim Rezaee, Houshang Rafatpanah.

**Writing – original draft:** Maryam Mahdifar, Arman Mosavat, Seyed Abdolrahim Rezaee, Houshang Rafatpanah.

**Writing – review & editing:** Arman Mosavat, Seyed Abdolrahim Rezaee.

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
