## [Decision Letter · Decision Letter 0]

28 Oct 2022

PGPH-D-22-01446

Immigrating and vicinity are not risk factors in the prevalence and transmission rate of HTLV-1: a survey in an endemic region of Iran and Afghan refugees

Dear Dr. Rezaee,

Thank you for submitting your manuscript to PLOS Global Public Health. After careful consideration, we feel that it has merit but does not fully meet PLOS Global Public Health’s publication criteria as it currently stands. Therefore, we invite you to submit a revised version of the manuscript that addresses the points raised during the review process.

Please address the comments provided by Reviewer 2. 

We look forward to receiving your revised manuscript.

Kind regards,

Vanessa Carels

Staff Editor

Journal Requirements:

3. Please provide separate figure files in .tif or .eps format only and remove any figures embedded in your manuscript file. Please also ensure that all files are under our size limit of 10MB.

4. In the online submission form, you indicated that "All datasets generated or analysed during the current study are included in this paper and are available from the corresponding author upon reasonable request". All PLOS journals now require all data underlying the findings described in their manuscript to be freely available to other researchers, either 1. In a public repository, 2. Within the manuscript itself, or 3. Uploaded as supplementary information.

5. Figs 1 and 2: please (a) provide a direct link to the base layer of the map (i.e., the country or region border shape) and ensure this is also included in the figure legend; and (b) provide a link to the terms of use / license information for the base layer image or shapefile. We cannot publish proprietary or copyrighted maps (e.g. Google Maps, Mapquest) and the terms of use for your map base layer must be compatible with our CC-BY 4.0 license. 

Additional Editor Comments (if provided):

Reviewers' comments:

Reviewer's Responses to Questions

**Comments to the Author**

1. Does this manuscript meet PLOS Global Public Health’s publication criteria? Is the manuscript technically sound, and do the data support the conclusions? The manuscript must describe methodologically and ethically rigorous research with conclusions that are appropriately drawn based on the data presented.

Reviewer #1: Yes

Reviewer #2: Yes

2. Has the statistical analysis been performed appropriately and rigorously?

Reviewer #1: Yes

Reviewer #2: Yes

3. Have the authors made all data underlying the findings in their manuscript fully available (please refer to the Data Availability Statement at the start of the manuscript PDF file)?

Reviewer #1: Yes

Reviewer #2: No

4. Is the manuscript presented in an intelligible fashion and written in standard English?

Reviewer #1: Yes

Reviewer #2: Yes

5. Review Comments to the Author

Reviewer #1: The manuscript entitle: Immigrating and vicinity are not risk factors in the prevalence and transmission rate of HTLV-1: a survey in an endemic region of Iran and Afghan refugees , describe the prevalence of HTLV infection betwwen Afghan refugees in Iran. The authors conclude that HTLV-1 is not endemic in Afghanistan, a war-stricken region with refugees distributed worldwideç and the land Silk Road has not been the route of HTLV-1 transmission to Northeastern Iran.

It is a important and relevant theme, since we had a lack about HTLV prevalence in Persian region. The methods, results, discussion and conclusion are very clearly , and the English langague is adequate.

Reviewer #2: The authors should rename the virus to Huma T-lymphotropic virus. Additionally, the authors stated that all samples from the two refugee camps were negative for HTLV-1 antibody, but refer that anti-HTLV-1 antibodies were detected in serum samples of five cases in the Fariman city group. Please, clarify this point. Where these subjects come from?

6. PLOS authors have the option to publish the peer review history of their article (what does this mean?). If published, this will include your full peer review and any attached files.

**Do you want your identity to be public for this peer review?** For information about this choice, including consent withdrawal, please see our Privacy Policy.

Reviewer #1: No

Reviewer #2: No

---

## [Decision Letter · Decision Letter 1]

14 Dec 2022

Immigrating and vicinity are not risk factors in the prevalence and transmission rate of HTLV-1: a survey in an endemic region of Iran and Afghan refugees

PGPH-D-22-01446R1

Dear Dr Rezaee,

We are pleased to inform you that your manuscript 'Immigrating and vicinity are not risk factors in the prevalence and transmission rate of HTLV-1: a survey in an endemic region of Iran and Afghan refugees' has been provisionally accepted for publication in PLOS Global Public Health.

Best regards,

Raquel Muñiz-Salazar, Ph.D.

Academic Editor

The authors have addressed all the comments and the manuscript is fine to be published

Reviewer Comments (if any, and for reference):

Reviewer's Responses to Questions

**Comments to the Author**

1. If the authors have adequately addressed your comments raised in a previous round of review and you feel that this manuscript is now acceptable for publication, you may indicate that here to bypass the “Comments to the Author” section, enter your conflict of interest statement in the “Confidential to Editor” section, and submit your "Accept" recommendation.

Reviewer #2: All comments have been addressed

2. Does this manuscript meet PLOS Global Public Health’s publication criteria? Is the manuscript technically sound, and do the data support the conclusions? The manuscript must describe methodologically and ethically rigorous research with conclusions that are appropriately drawn based on the data presented.

Reviewer #2: Yes

3. Has the statistical analysis been performed appropriately and rigorously?

Reviewer #2: N/A

4. Have the authors made all data underlying the findings in their manuscript fully available (please refer to the Data Availability Statement at the start of the manuscript PDF file)?

Reviewer #2: Yes

5. Is the manuscript presented in an intelligible fashion and written in standard English?

Reviewer #2: Yes

6. Review Comments to the Author

Reviewer #2: The authors have addressed all the comments and the manuscript is fine to be published

7. PLOS authors have the option to publish the peer review history of their article (what does this mean?). If published, this will include your full peer review and any attached files.

**Do you want your identity to be public for this peer review?** For information about this choice, including consent withdrawal, please see our Privacy Policy.

Reviewer #2: No
